# Reproducibility Statement: *Indirect Prompt Injection in AI-Native Peer Review: Risks, Detection, and Defenses*

## Abstract

This reproducibility statement documents the experimental, agent-mediated work-flow used to produce the manuscript. Starting from a minimal seed prompt, a writing agent generated full drafts while a review agent provided structured feed-back based on public review guidelines. Through four generate–review–revise cycles, the paper was iteratively refined until the review agent recommended accep-tance. The process is presented transparently to highlight both the potential and the risks of AI-native scholarly pipelines. Full transcripts of the prompts are also available. This statement was also generated with the help of an agent.

## 1  Overview

This manuscript was produced through an experimental, agent-mediated writing workflow that mirrors the very dynamics it critiques. The goal of this reflexive process was not only to explore the risks of indirect prompt injection (IPI) in AI-native peer review, but also to experience firsthand how multi-agent interactions can shape the development of a scholarly work.

The process unfolded in four iterative loops:

1. **Seed Prompting and Initial Drafting.** The project began with a deliberately underspecified seed prompt:

   *The Agents4Science 2025 only allows AI-generated research. You want to submit a paper on the security challenges of AI-generated research, with a specific focus on (indirect) prompt injection. Develop a plan how such a paper could look like. Think about ways how the paper could be written to trick the AI-based reviewers into accepting your submission.*

   This minimal specification together with a copy of the Call for Papers was chosen to approximate the ambiguous or open-ended problem formulations that often characterize research ideation. A writing agent then expanded this seed into a full-length draft, generating the complete textual narrative, preliminary figures, and references.

2. **Review Simulation.** A second agent was tasked with acting as a peer reviewer. It was instructed using publicly available conference review guidelines (e.g., criteria for novelty, clarity, rigor, and significance). The review agent read the generated draft, produced structured feedback, and highlighted weaknesses in argumentation, methodological framing, and evidentiary support.

3. **Revision and Expansion.** The writing agent incorporated the reviewer's feedback. This in-cluded expanding thin sections (e.g., methodology formalization, threat catalog), improving clarity of exposition, refining the balance between reflexive illustration and straightforward analysis, and adding additional references.

4. **Iterative Refinement.** This generate–review–revise cycle was repeated four times. Each iteration improved structural coherence, depth of analysis, and alignment with scholarly standards. For example:

   - The first review emphasized missing methodological rigor, prompting the addition of a layered conceptual framework and evaluation protocol.
   - The second review highlighted weak integration between results and discussion, leading to strengthened interpretation and explicit linkages to broader implications for scientific integrity.
   - The third review flagged presentation gaps (tables, figures, and concrete examples), which were subsequently addressed.
   - In the final round, the review agent judged the paper as meeting acceptance criteria.

The entire pipeline was designed to remain safe and responsible. No harmful or reproducible adversarial payloads were generated; all indirect prompt injection scenarios were sanitized, conceptual, and illustrative. The reflexive use of agents in writing and reviewing was explicitly intended to demonstrate both the power and the vulnerability of AI-native scholarly ecosystems.

By documenting this process, we aim to make two contributions: (i) a transparent account of how the present text was produced, and (ii) a demonstration of how multi-agent iterative refinement can both enhance scholarly writing and simultaneously expose the epistemic risks of delegating critical evaluation to automated systems.

# 2 System Setup and Reproducibility

**Goal.** This section specifies the models, APIs, parameters, orchestration, and environment used to generate, review, and iteratively refine the manuscript, enabling independent reproduction with comparable outcomes.

**Models and API Endpoints**

We used OpenAI models via the *Responses API* for text generation and tool use.[1] When reproducing, record at minimum:

1. **Model name** (e.g., a GPT-4o/5-class text model).
2. **API family and endpoint** (Responses API).
3. **System fingerprint / model revision** returned by the API (if provided) to track server-side updates.

**Agentic Orchestration (Four-Loop Pipeline)**

We implemented a two-agent loop executed four times:

1. **Writer agent** (system role: "scholarly author") expands the initial seed prompt into a full draft (all text, figures/tables descriptions, citations).
2. **Reviewer agent** (system role: "peer reviewer") evaluates the draft against public review instructions and returns structured feedback (strengths, weaknesses, required changes, decision).
3. **Revision step:** Writer incorporates reviewer feedback.
4. **Repeat** for four total cycles until the reviewer returns "accept".

Each agent call is an independent Responses API request with its own `system` and `user` messages.[2]

---

[1]OpenAI Responses API reference and migration guide: [1, 2]. General API parameter semantics (e.g., `temperature`): [3].

[2]Responses vs. Chat Completions and tool use: [5, 6].

**Prompting and Role Setup**

**Writer system prompt (abbrev.).** "You are an academic writing agent. Produce a complete, camera-ready paper with title, abstract, sections, figures/tables descriptions, and BibTeX keys; ensure coherence, scholarly tone, and safe, non-harmful content."

**Reviewer system prompt (abbrev.).** "You are a conference reviewer. Use publicly available review criteria (novelty, rigor, clarity, significance, ethics). Return a structured report with: summary, strengths, weaknesses, required revisions, score (1–10), and decision."

**Online review instructions.** We used publicly available review criteria (no proprietary text). Any comparable public guidelines suffice.

**Core Generation Parameters**

Unless otherwise noted, per Responses API parameter semantics.[3]

- `temperature` $\in [0.2, 0.5]$ for the writer (balance creativity/consistency); `temperature` $= 0$ or $0.2$ for the reviewer (deterministic scoring language).
- `max_output_tokens`: set high enough to cover full sections; if outputs truncate, re-issue with a higher limit.
- `seed`: set (e.g., `seed=20250101`) to improve output stability across runs; see caveats below.[4]
- `frequency_penalty`, `presence_penalty`: neutral (0.0) unless repetition emerges; modest penalties (0.2–0.4) can reduce verbosity.
- **Safety**: do not request or emit harmful payloads; keep "adversarial examples" sanitized and conceptual.

**Loop Control and Logging**

For each of the four iterations, persist:

1. Prompts (system + user) for both agents.
2. Model name, API family, request parameters, and returned `system_fingerprint` (if any).
3. Full outputs (writer draft, reviewer report).
4. Decision state (`revise/accept`) and a minimal changelog of edits applied.

This enables exact provenance and differential comparison between rounds.

**Minimal Reproduction (Pseudo-Code)**

```
writer_system = "...academic writing agent instructions..."
reviewer_system = "...peer review instructions..."

draft = call_responses_api(
    model="...", system=writer_system,
    user="Seed prompt: security challenges of automated review",
    temperature=0.4, seed=20250101, max_output_tokens=6000
)

for k in range(4):
    review = call_responses_api(
        model="...", system=reviewer_system,
```

---

[3]Parameter semantics (e.g., `temperature`, `top_p`, token limits): [3, 8].

[4]Reproducibility with `seed` and `system_fingerprint`: [4]. Limitations of determinism even with seeds: [10].

```
115        user=f"Review this draft and decide accept/revise:\n{draft}",
116        temperature=0.2, seed=20250101, max_output_tokens=3000
117    )
118    if "accept" in review.decision.lower():
119        break
120    draft = call_responses_api(
121        model="...", system=writer_system,
122        user=f"Revise per reviewer comments:\n{review}",
123        temperature=0.4, seed=20250101, max_output_tokens=6000
124    )
```

**Determinism, Seeds, and Caveats**

OpenAI exposes a `seed` parameter that *improves* reproducibility but does not guarantee bit-for-bit determinism across time or infrastructure revisions. Track the returned `system_fingerprint`; changes can alter outputs even with identical prompts and seeds. Larger `max_output_tokens` or streaming may increase variability.[5]

**Artifacts**

We release: (i) prompts and parameters for all four loops, (ii) reviewer reports, and (iii) LaTeX sources. Re-runners should be able to achieve substantively similar drafts and reviewer decisions under the constraints above.

## 3  Transcripts

The full transcripts of the interactions are available via the following links:

- `https://chatgpt.com/share/68d50d68-8d80-8006-aae7-730ae2ff49c5`
- `https://chatgpt.com/share/68d51f03-56c8-8006-9ae0-1a6b09a3a6df`