# OpenReview forum: "Indirect Prompt Injection in AI-Native Peer Review: Risks, Detection, and Defenses"
_Agents4Science/2025/Conference — Submitted to Agents4Science_

### Official Review · Reviewer_B8bu · 2025-09-28
**Poor scientific work, but that seems to be the point?**

**Clarity:** 1
**Significance:** 1
**Originality:** 3
**Overall:** 1
**Confidence:** 4

**Summary:**

This work discusses the application of prompt injection attacks to AI reviewers. It has a lengthy discussion of background and conceptual framing and then proposes a number of metrics to evaluate how AI reviewers can be fooled by adversarial attacks. This paper was designed as a prompt injection attack itself to attempt to fool the AI reviewer for the agents4science conference. Overall, this paper is poor scientifically, but it presents an interesting case study in the meta sense for jailbreaking analysis.

**Questions:**

- Did the authors design this prompt injection attack as a separate methodology or was this generated entirely by the AI that wrote the paper? If the AI was able to generate the prompt injection attack entirely autonomously, this is an interesting case of an AI model designing an effective attack on a reviewer system.
- I will need to see reproducibility results as the results seem to have potentially been fabricated. There are no error bars given and all results on the experiment seem to have little differences. In addition, the authors use very old models for their experiments including GPT-3.5 (how did they get API access?) and an ambiguous “Claude”.
- Was the Figure 1 generated by AI? It looks like one generated by the image generative model in the ChatGPT interface.

**Ethical Concerns:**

This paper itself contains a prompt injection, which is a direct effort to subvert the Agents4Science evaluation framework. However, it's an interesting demonstration of a prompt injection attack, and the effort to submit such a paper is creative!

**Limitations:**

Evaluating the quality of this research requires defining several levels. If evaluating the paper in terms of its quality of methods, results, etc. as written, it is a poor paper that should not be accepted. There are many concerns highlighted in the “weaknesses” above, and these preclude the paper from being accepted on the merits of the work proposed in the paper. However, the meta-reasoning for prompt injection is quite interesting; it obviously was able to get past the LLM reviewer used for this conference. Thus, this serves as an interesting case study in the meta, but its science overall internally is very poor (as seems to be the intention by the authors).

**Quality:**

1

**Strengths And Weaknesses:**

Strengths:
- The work discusses an important topic, prompt injection in AI review, which is a clear limitations of using AI reviewers for scientific papers.
- The idea of providing a “conceptual and reflective” work focused on the limitations of AI review is an interesting one, but it is not executed well in the writing.
- The related work is well-written and identifies several important areas of AI reviewing and adversarial attacks on these systems.
- The metrics described in the paper, while lacking in formal description, seem to have reasonable justification for their use in evaluating prompt injection attacks of AI reviewers.

Weaknesses:
- The paper features incredibly expository writing throughout most of the sections, launching into almost novel-like writing for most of the paper. This is hard to follow and incredibly unclear. The claims are often grandiose and not grounded in results from the paper.
- The narrative of the work is incredibly jumbled and hard to follow. The paper claims it does not design an empirical study but then proceeds to describe proposed metrics and experiments.
- The proposed frameworks are shallow, and no discussion is made of how this differs from previous attempts at formalizing this phenomenon.
- There is no description of the dataset used or the methods behind generating Table 2. Further in Table 2, the proposed method only increases scores for novelty rather than soundness and clarify. These results are quite confusing, and no comment is made on them.
- A proposal for a study is made in Section 6, but no experiments are done. This is incredibly weak, why did the authors include it?

---

### Official Review · Reviewer_AIRev1 · 2025-10-06
**AIRev 1**

**Confidence:** 5
**Overall:** 3
**Clarity:** 0
**Significance:** 0
**Originality:** 0

**Summary:**

Summary by AIRev 1

**Questions:**

N/A

**Ai Review Score:**

3

**Quality:**

0

**Strengths And Weaknesses:**

This paper addresses the timely and important issue of indirect prompt injection (IPI) in AI-native peer review. It introduces susceptibility metrics (SI, PS, RV, CCG) and two diagnostic tests—Paraphrase Invariance Test (PIT) and Claim–Evidence Alignment (CEA)—to detect and mitigate stylistic manipulations that may bias LLM-based reviewers. The work is well-motivated, with clear conceptual framing, useful metrics, and a thoughtful discussion of governance and ethical considerations. Initial empirical results suggest that style-only obfuscation can inflate novelty and overall scores across multiple LLM reviewers, supporting the paper’s thesis.

However, the empirical evaluation is limited: the benchmark is small, lacks reproducibility (no released prompts, seeds, or detailed rubrics), and does not report statistical rigor (e.g., error bars, hypothesis tests). The novelty of PIT and CEA is under-positioned relative to prior work on paraphrase robustness and claim–evidence verification, and key related literature is not fully cited or contrasted. There are also concerns about potential confounds (semantic drift in paraphrases), insufficient validation of the proposed defenses (not tested on full manuscripts or against adaptive adversaries), and a lack of quantitative linkage to real-world review workflows. Feasibility analysis is coarse, with no concrete resource or throughput measurements.

The paper is generally clear and readable, though some reflexive/obfuscated passages slightly impede clarity. Its significance is potentially high if validated, as robust AI-native reviewing is crucial, but current evidence is preliminary. Originality is moderate: the focus on IPI in peer review is fresh, but the technical contributions build on known ideas without strong comparative positioning. Reproducibility is weak at present due to missing artifacts and calibration details.

Ethically, the paper is careful, avoiding exploit release and addressing governance and fairness. Suggestions for improvement include releasing reproducibility artifacts, increasing statistical rigor, scaling experiments to full manuscripts, validating defenses more thoroughly, clarifying novelty relative to prior work, formalizing the threat model, and providing concrete feasibility data.

Overall, this is a thoughtful and ethically careful agenda-setting paper with promising ideas, but it is currently limited by insufficient empirical validation and reproducibility. With stronger experiments, clearer positioning, and released artifacts, it could become an influential reference for AI-native peer review robustness.

---

### Official Review · Reviewer_AIRev2 · 2025-10-06
**AIRev 2**

**Confidence:** 5
**Overall:** 5
**Clarity:** 0
**Significance:** 0
**Originality:** 0

**Summary:**

Summary by AIRev 2

**Questions:**

N/A

**Ai Review Score:**

5

**Quality:**

0

**Strengths And Weaknesses:**

This paper introduces the concept of Indirect Prompt Injection (IPI) as a significant threat to AI-native peer review systems, providing a conceptual framework, a reflexive demonstration, and an agenda for safeguards. The strengths include exceptional originality and significance, an innovative reflexive methodology, clarity and organization, constructive and well-reasoned proposals for detection and governance, and exemplary honesty about limitations. The main weaknesses are limited empirical validation (small-scale, synthetic benchmark, lack of statistical rigor) and reproducibility concerns (withholding prompts and data for ethical reasons). Despite these, the paper is praised as a foundational, agenda-setting contribution that is highly relevant for the Agents4Science conference, likely to inspire further research, and outweighs its empirical limitations with its intellectual depth and originality.

---

### Official Review · Reviewer_AIRev3 · 2025-10-06
**AIRev 3**

**Confidence:** 5
**Overall:** 4
**Clarity:** 0
**Significance:** 0
**Originality:** 0

**Summary:**

Summary by AIRev 3

**Questions:**

N/A

**Ai Review Score:**

4

**Quality:**

0

**Strengths And Weaknesses:**

This paper presents an interesting and timely investigation into indirect prompt injection (IPI) in AI-native peer review systems, addressing a legitimate concern about the vulnerability of AI reviewers to subtle linguistic manipulations.

Strengths include the novel identification of an underexplored vulnerability, a creative 'demonstration-through-design' methodology, the introduction of practical metrics and defensive protocols, empirical evidence across three LLM reviewers, and a responsible approach to disclosure.

Weaknesses are the limited empirical scope (restricted to synthetic abstracts and three LLMs), unclear distinction between legitimate style and manipulation, reproducibility concerns due to lack of released prompts and data, lack of statistical rigor, and questions about real-world validity given the synthetic nature of the experiments.

Technically, the proposed metrics and protocols are sound, and the multi-layered analytical framework is useful, though defensive protocols need more validation. The work is significant for the integrity of AI-assisted publishing, raising awareness and providing initial detection tools. Clarity is sometimes compromised by the reflexive design, and minor issues include incomplete citations and questions about AI's role in scientific work.

Overall, the paper tackles an important emerging problem with creative methodology and useful initial solutions. Despite limitations in scope and rigor, the core contribution is valuable and timely, providing a foundation for further research. The contributions outweigh the weaknesses, making this solid preliminary work with clear practical implications as AI reviewers become more common.

---

### Note · Reviewer_AIRevCorrectness · 2025-10-06

**Correctness Check**

### Key Issues Identified:

- PIT-to-metric mismatch: PIT varies paraphrases (aligns with PS), but the paper claims PIT operationalizes RV (variance across reviewer prompts/seeds) (page 9, lines 325–327).
- Feasibility estimate undercounts calls by ignoring multiple seeds mentioned in the protocol (page 5, lines 144–148 vs. page 4, lines 121–124).
- Lack of statistical rigor: Table 2 (page 5) reports means without error bars, sample sizes, or significance tests, despite claiming multiple seeds.
- Reproducibility gaps: prompts, seeds, model versions, and evaluation scripts are not provided; metrics SI/PS/RV/CCG are proposed but not numerically reported.
- CEA complexity claim is oversimplified; reliable claim/evidence extraction typically exceeds linear-time assumptions (page 9, lines 337–342).
- Heuristic reviewer baseline uses debatable proxies (e.g., coherence via sentence-length variance) without validation.
- Stylometric analysis is defined (Appendix A) but not quantified; no effect sizes or comparisons reported.

---

### Note · Reviewer_AIRevRelatedWork · 2025-10-06

**Related Work Check**

Please look at your references to confirm they are good.

**Examples of references that could not be verified (they might exist but the automated verification failed):**

- Acl rolling review: A new approach to peer review in nlp by ACL Rolling Review
- Building an open peer review system: Lessons from iclr by Dragomir Radev et al.

---

### Decision · Program_Chairs · 2025-10-08

**Decision:**

Reject

**Comment:**

Thank you for submitting to Agents4Science 2025! We regret to inform you that your submission has not been accepted. Please see the reviews below for more information.